# Field Implementation of Forecasting Models for Predicting Nursery Mortality in a Midwestern US Swine Production System

**DOI:** 10.3390/ani13152412

**Published:** 2023-07-26

**Authors:** Edison S. Magalhaes, Danyang Zhang, Chong Wang, Pete Thomas, Cesar A. A. Moura, Derald J. Holtkamp, Giovani Trevisan, Christopher Rademacher, Gustavo S. Silva, Daniel C. L. Linhares

**Affiliations:** 1Department of Veterinary Diagnostic and Production Animal Medicine, College of Veterinary Medicine, Iowa State University, Ames, IA 50011, USA; 2Department of Statistics, College of Liberal Arts and Sciences, Iowa State University, Ames, IA 50011, USA; 3Iowa Select Farms, Iowa Falls, IA 50126, USA

**Keywords:** swine, mortality, data-wrangling, forecasting, machine-learning

## Abstract

**Simple Summary:**

Swine nursery mortality is highly impacted by the pre-weaning performance of the piglets. Even though the importance of the pre-weaning phase on the downstream post-weaning performance is acknowledged, predictive modeling has yet to be described in the swine industry to predict the downstream nursery performance of groups of pigs based on their previous pre-weaning phase. One obstacle to building such predictive models is that pieces of information concerning the factors impacting swine mortality are collected with separate record-keeping programs and stored in unconnected databases, creating multiple unutilized data stream clusters. Thus, in this study, we described the process of building a data-wrangling pipeline that automatically integrates diverse and dispersed data streams collected from one swine production company, creating then a master table that was utilized to predict the mortality of groups of pigs during the nursery phase.

**Abstract:**

The performance of five forecasting models was investigated for predicting nursery mortality using the master table built for 3242 groups of pigs (~13 million animals) and 42 variables, which concerned the pre-weaning phase of production and conditions at placement in growing sites. After training and testing each model’s performance through cross-validation, the model with the best overall prediction results was the Support Vector Machine model in terms of Root Mean Squared Error (RMSE = 0.406), Mean Absolute Error (MAE = 0.284), and Coefficient of Determination (R^2^ = 0.731). Subsequently, the forecasting performance of the SVM model was tested on a new dataset containing 72 new groups, simulating ongoing and near real-time forecasting analysis. Despite a decrease in R^2^ values on the new dataset (R^2^ = 0.554), the model demonstrated high accuracy (77.78%) for predicting groups with high (>5%) or low (<5%) nursery mortality. This study demonstrated the capability of forecasting models to predict the nursery mortality of commercial groups of pigs using pre-weaning information and stocking condition variables collected post-placement in nursery sites.

## 1. Introduction

The abundance of diverse and large-scale data streams often challenges the implementation of precision animal agriculture in livestock, which requires a multifaceted data-wrangling approach to investigate this complex livestock “big data” [1]. Using data management techniques and machine-learning models on these data can overcome its complexity for analytical purposes, such as forecasting. Although forecasting analysis in the livestock realm is acknowledged [2,3], this application has not yet been reported in the swine industry for mortality rate. Swine post-weaning mortality is a key performance indicator (KPI) utilized to measure the sustainability of swine production systems [4,5], divided into nursery and finisher mortality. Swine nursery mortality refers to the mortality of pigs in the first 5–8 weeks of the overall post-weaning phase (approximately 5.5 months), accounting for a large portion of the overall post-weaning mortality [6].

Information concerning the risk factors for swine mortality is routinely collected, such as health, environment, productivity, and infrastructure. However, integrating and merging these data streams is necessary for its collective utilization for targeting prediction or risk factor analyses. The development of means for data integration and analysis under field conditions allows the implementation of such data analysis approaches, as reported in previous studies [7,8,9,10]. Therefore, the objective of this study was to develop a data-wrangling pipeline within one swine production system to integrate and manage multiple data streams, enabling automated and near real-time data consolidation. Furthermore, the performance of multiple forecasting models was assessed on historical data, and the best model was tested on new data to predict the nursery mortality of prospective closeouts.

## 2. Materials and Methods

### 2.1. Overview and Study Design

This study utilized field data from a large U.S. swine production system in the Midwestern region. A total of six different and disconnected data streams related to 3242 groups of marketed pigs (over 13 million animals) slaughtered over three years, here referred to as closeouts, were collected for the analyses. The retrospective performance of both the pre- and post-weaning phases of production were imported and integrated into the respective closeouts’ information, constructing a dataset (also known as the master table) containing breeding-to-market historical information for each closeout. The pre-weaning phase variables and stocking conditions data in this master table were utilized as predictors to forecast the downstream post-weaning mortality of each closeout on their initial 60 days in the post-weaning phase (nursery mortality), as demonstrated in Figure 1.

Closeouts were defined as the groups of pigs originating from the company’s breeding herds. The pigs remained in the breeding herd until weaning at approximately 21 days of age. Following weaning, pigs were placed on feed at growing sites where the groups remained for around 5.5 months. The groups were managed all-in-all-out, meaning another group of pigs could only start once all the pigs from the previous groups had been marketed. The mortality of each closeout during the nursery phase was defined as the outcome variable of analysis in this study and was calculated as the following: (number of pigs at placement − number of pigs 60 days post placement) ÷ number of pigs at placement.

Closeouts originating from a single sow farm would have information concerning the performance of that breeding herd on the designated week. The productivity parameters assigned to the downstream weaned group represent the retrospective performance of that batch of pigs from farrow-to-wean, while the pre-farrow information (e.g., abortion rate) represents the reproductive performance on that farm on the week of the weaning event.

When a group of pigs originated from multiple sow farms (e.g., 2000 pigs placed in a growing site may have received 1000 pigs from two different sow farms), the variables concerning the pre-weaning phase for that specific group would be calculated by using a weighted average for the continuous independent variables, and the mixed classification was used for the whole group for disease classification statuses.

SAS^®^ Version 9.4 (SAS Institute, Inc., Cary, NC, USA) was utilized to build data-wrangling pipeline algorithms, thus automating the processes of importing, managing, cleaning, and integrating the data streams. The integration of the six data streams resulted in a final master table for the 3242 closeouts that were utilized for comparing the performance of five different regression and machine-learning models for forecasting swine nursery mortality. After this step, the model with the best forecasting performance was utilized on a new dataset to validate the forecasting capability on prospective data, simulating ongoing near real-time forecasting.

### 2.2. Data-Wrangling Pipeline

The six different data streams available for the development of the master table were: (1) pre-weaning phase (i.e., breeding herd) productivity and health data; (2) post-weaning phase (i.e., growing phase) productivity data; (3) closeouts’ health status reports; (4) pig transportation records; (5) stocking conditions reports; and (6) management procedure records. The SAS algorithms developed in this study used a similar methodology to that described by Magalhaes et al. (2022) [10], where the processes of matching and merging different data streams were conducted based on an identifier (time and location of events) and through the development of PROC Statements algorithms (PROC MERGE, PROC SET, PROC SQL, PROC SORT, PROC UNIVARIATE, and PROC FREQ). The swine production system provided access to the aforementioned data, where a data workflow was developed using Microsoft Power Automate (Microsoft Corporation, Redmond, WA, USA) and SAS to automate the data-wrangling processes in this study. Once the master table was built, the dataset contained information for 3242 closeouts of pigs, originating from 42 breeding herd sources and weaned into 529 different growing sites. The information from each of the six data streams was matched and merged to each respective closeout of pigs marketed in this study period (i.e., each closeout’s historical data from breeding-to-market).

### 2.3. Comparing Forecasting Models Based on Training Data

The initial step after completing the master table was to select the breeding herd variables from the pre-weaning phase of production and parameters that represent the stocking conditions of the weaned groups into growing sites (i.e., characteristics at placement). Among all variables in the master table, 42 parameters were utilized as predictors in the forecasting analyses (Table 1). The nursery mortality was log-transformed after verifying that its distribution was not normal, thus, utilizing the log-mortality as the response variable. The classes of each categorical variable included in the model are described in Table 2. 

The 42 variables included in the forecasting analysis were selected based on their potential as factors related to the quality of weaned pigs (i.e., health status and productivity performance) and the overall conditions at placement in growing sites (i.e., infrastructural and management factors). Also, only variables that were provided in the master table at the moment when weaned pigs were placed into growing sites were included in the model.

To forecast the log-mortality, five models were investigated: multiple linear regression model (MLR), LASSO regression, support vector machine (SVM), neural network (NNet), and random forest (RF). The evaluation criteria for each forecasting model included Root Mean Squared Error (RMSE), Mean Absolute Error (MAE), and Coefficient of Determination (R^2^). Using the R package ‘caret’ [11], and specifically the ‘train’ function, the optimal parameters of LASSO regression, SVM, and NNet were selected based on the smallest RMSE by doing three repetitions of 5-fold cross-validation, and the optimal parameters of RF were selected based on the smallest out-of-bag (OOB) error. 

In order to evaluate the prediction performance of each forecasting model, a leave-one-out cross-validation was performed, where, for each record, the training set was the dataset excluding that record. The trained model was then used to predict the log-mortality of the excluded record. The best model was selected based on higher R^2^ and lowest RMSE and MAE values.

### 2.4. Performance of the Selected Model on Independent Validation Data

After comparing the performance of the different forecasting models on the retrospective dataset of 3242 groups, which refers to groups stocked into nursery sites between week 29 of 2019 through week 5 of 2022 (i.e., marketed between January 2020 to August 2022), a new dataset containing 72 new closeouts weaned into nursery sites between weeks 6 and 12 of 2022 (i.e., marketed between August and September of 2022) was obtained through the data-wrangling pipeline. The forecasting model was then utilized on this naïve data to predict the nursery mortality of the groups, and the forecasting performance of the selected model was measured using the same metric of the same step (R^2^, RMSE, and MAE). Also, the predicted vs. actual nursery mortality values were classified into relatively “high nursery mortality” (>5%) or “low nursery mortality” (<5%) groups, as the company providing the data used the same classification as their target mortality values. The performance of the SVM model on accurately predicting closeouts with high or low nursery mortality was assessed in terms of accuracy (Ac), sensitivity (Se), Specificity (Sp), positive predicted value (PPV), and negative predicted value (NPV), calculated based on the difference between the predicted vs. actual mortality of the 72 groups.

## 3. Results

### 3.1. Data-Wrangling Pipeline

When assessing data completeness for the 3242 groups, a total of 93 closeouts (2.87%) were excluded due to a lack of information for all the characteristics included in the master table, resulting in a final dataset composed of 3149 closeouts and 42 explanatory variables to be used in the forecasting analyses. 

### 3.2. Comparing Forecasting Models

The overall performance for all forecasting models is reported in Table 2. Notably, the machine learning models performed better than the regression models, where the RF and SVM models demonstrated the best overall prediction performance, similar to other livestock-related studies comparing the performance of multiple forecasting models [3,12,13]. Furthermore, the SVM outperformed the other models (Table 3) measured in terms of R^2^ (0.731) and lower errors measured by RMSE (0.406) and MAE (0.284).

Thereafter, the predicted values for each closeout using the SVM model were averaged by week for the data collected in this study (Figure 2), where it was observed that the SVM predicted values were underestimated compared to the actual nursery mortality values of the closeouts. Despite this, both the average weekly predicted and actual mortalities followed similar seasonal trends over time, which can be explained by the seasonal activity of major diseases impacting the swine industry [14,15].

### 3.3. Performance of the Selected Model

Identified as the superior model, SVM was prospectively applied to new data consisting of 72 closeouts (Figure 3), representing one month of closeouts, to predict the nursery mortality of the new groups. The overall forecasting performance of the SVM model was lower than the training database’s performance on the cross-validation procedure (R^2^ = 0.554 and 0.731, respectively). However, even though the prediction of mortality in new groups was already simulated in the training database during the cross-validation procedure, the prediction performance was inferior when applying the same model to a smaller sample of a prospective dataset. It is important to note that the training step was conducted in a much larger dataset, while the testing of the SVM model was conducted in a smaller dataset.

Despite the SVM’s decreased performance on naïve data when categorizing both predicted and actual nursery mortality of the 72 closeouts into high (>5%) or low (<5%) nursery mortality, a high accuracy value (77.78%) was observed for the SVM on correctly predicting the closeouts as high or low nursery mortality. Also, we observed that most of the groups are located in the positive diagonal axis of the chart, which is the desired area in terms of prediction (Figure 3). 

The values for sensitivity (62.16%), specificity (94.29%), positive predicted value (92.00%), and negative predicted value (70.21%) also demonstrated an acceptable prediction performance, especially for precisely predicting groups with “high nursery mortality” rates (i.e., at high risk). Overall, the SVM model accurately predicted 62.16% of the closeouts with relatively “high nursery mortality” and 94.29% with relatively low mortality. In other words, even though the SVM model did not predict all groups that had “high nursery mortality” as high (false negatives), the model had a high positive predicted value, indicating that 92.00% of the closeouts predicted as “high nursery mortality” were observed as actually high. 

For the categorical variables (*n* = 7) included in the forecasting model, when comparing the frequency distribution between the number of closeouts with high and low mortality groups compared to their respective predicted values (Figure 4), the forecasting model overestimated the number of groups with low predicted mortality (i.e., right-side transparent bars are longer than the right-side solid bars). On the other hand, the forecasting model underestimated the actual number of closeouts with “high nursery mortality” for all classes of the categorical variables illustrated (i.e., left-side transparent bars are shorter than the left-side solid bars). 

Notably, for specific classes within the categorical variables (e.g., “Pig med.—Tulathromycin” or “PRRS Status—Epidemic”), the proportion of groups predicted as “high nursery mortality” were higher than the number of groups predicted as “low nursery mortality”. This hypothesis is supported by the common knowledge that PRRS infections in breeding herds generate downstream PRRS-epidemic weaned pigs [10,16,17,18,19,20], which are expected to be more challenged throughout the post-weaning phase. Also, the use of tulathromycin to treat piglets in breeding herds indicates that health-challenged pigs were weaned, as this is a frequently prescribed antibiotic in swine due to its ability to modulate the immune system, as well as an effective treatment against key respiratory diseases [21].

On the other hand, for some factors such as “MhP—Negative” or “Pig med.—None”, the largest proportion of the groups of pigs were predicted as low nursery mortality groups, which can be explained by the fact that *M. hyopneumoniae* infection in weaned pigs can increase growing pig mortality [22], thus, negative pigs are expected to have higher survivability. Also, the presence of groups of weaned pigs that were not treated with medication during the lactation can indicate groups with higher quality that did not need this procedure.

Altogether, the results demonstrated in Figure 4 indicate the influence of specific factors on the overall prediction. However, this study was not designed to investigate the influence of these specific factors on nursery mortality, as this type of approach requires a causal inference analysis [23,24], which was not the scope of this study.

## 4. Discussion

The algorithms developed in this study for the data-wrangling pipeline allowed the integration of information previously stored independently and underutilized for analysis purposes, combining the dispersed predictors in multiple data streams into a single master table. This approach of combining multiple data streams to investigate post-weaning performance was previously described in other studies [7,8,25,26,27].

Multiple machine learning and regression algorithms were applied to the master table to compare their forecasting performance in predicting swine nursery mortality. Also, other studies in the livestock realm demonstrated the application of similar models for predicting important KPIs of productivity [2,3,28,29,30,31,32,33]. 

Assuming that the swine production system maintains the format of the data streams utilized to build the master table over time, the algorithms can be utilized to integrate and prepare new incoming information for prospective analyses, including forecasting and causal inference, as it is seen that incompatibility of data streams is one of the major challenges in data integration [34].

The results of both the data-wrangling pipeline procedure and the forecasting models’ comparison allowed the training of the best model on retrospective data and further testing on new data, simulating the ongoing application of forecasting models on future data, in other words, utilizing the pre-weaning phase and stocking condition variables to predict the future mortality of closeouts.

The algorithms developed in this study can support swine practitioners in their decision-making process to strategically allocate resources (or not) for groups with predicted high nursery mortality. Notably, the predictive performance of the models refers specifically to the dataset collected in this study and to the time analyzed. In other words, the performance may change over time within this company as swine nursery mortality is impacted by multifactorial components that are dynamically interacting over time and period [4,5,10], limiting the external validity of this study to other field conditions.

Although there is an opportunity for improving the prediction of the exact values of nursery mortality (i.e., continuous outcome), there is a trade-off between prediction error and the utility of the predicted value when using binary vs. continuous outcome. For example, more relevance was given by the production system in this study to identify relatively high nursery mortality groups instead of predicting their exact mortality values.

The lower sensitivity results of this study can be explained by limiting the inclusion of predictor variables related only to the pre-weaning phase and conditions of weaned pigs at placement in growing sites (stocking conditions variables), as post-weaning infectious and non-infectious factors are likely to increase swine mortality as well [5]. However, as the goal of this study was to forecast nursery mortality at the beginning of the post-weaning phase (at placement), a trade-off of losing accuracy in terms of prediction but allowing early intervention is expected.

On the other hand, the model demonstrated a high performance when predicting groups that would have high nursery mortality (high positive predicted value), thus indicating that sow farm variables related to the quality of the piglets at weaning can drive their mortality throughout the post-weaning phase as demonstrated by other authors [35,36,37,38].

## 5. Conclusions

Forecasting swine nursery mortality can support decision-makers in allocating resources or interventions toward precision swine health and productivity management. This study demonstrated the capability of building system-specific algorithms that allows the development of an automated data-wrangling pipeline, which enables ongoing and near real-time forecasting. Also, this study demonstrated the ability to utilize breeding herd characteristics and data concerning the stocking conditions of weaned pigs placed in nursery sites as predictors for forecasting nursery mortality. Despite the overall acceptable performance for predicting groups at high nursery mortality risk, there is an opportunity for improving the model’s performance by including more predictors and other machine-learning models.

## Figures and Tables

**Figure 1 animals-13-02412-f001:**
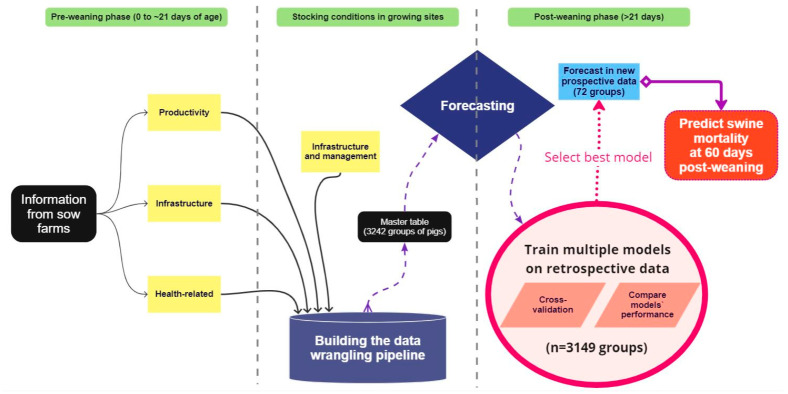
Flow chart explaining the process of integrating different data streams into a single master table to be used on the forecasting analyses. A data-wrangling pipeline was built to integrate information concerning the performance of groups of weaned pigs transferred to growing sites and their conditions at placement into a single master table. This table was utilized to train forecasting models on predicting the mortality of the weaned groups throughout the initial 60 days in the post-weaning phase. The model with the best performance was then applied to forecast the mortality of prospective new groups of weaned pigs.

**Figure 2 animals-13-02412-f002:**
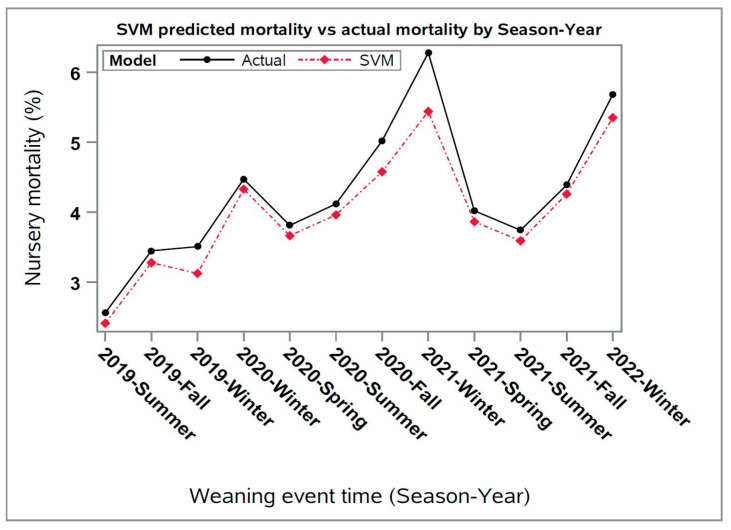
Average predicted nursery mortality versus actual mortality over season-year for Support Vector Machine (SVM) forecasting model using the results of the cross-validation step (3149 closeouts). Season-Year corresponds to the time during the study period when the pigs were weaned.

**Figure 3 animals-13-02412-f003:**
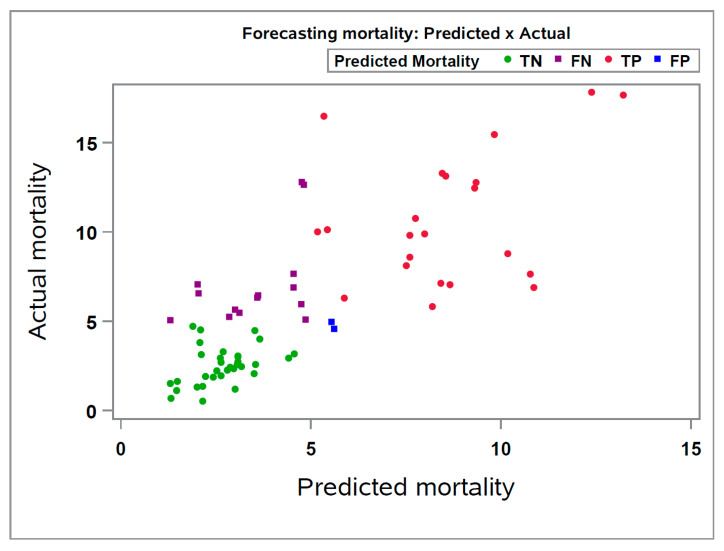
Correlation plot between the observed and predicted nursery mortality using the SVM model on 72 new closeouts. True Negative (TN): Predicted as Low mortality (<5%) and actual was Low (<5%); False Negative (FN): Predicted as Low mortality (<5%) and actual was High (>5%); True Positive (TP): Predicted as High mortality (>5%) and actual was High (>5%); False Positive (FP): Predicted as High mortality (>5%) and actual was Low (<5%).

**Figure 4 animals-13-02412-f004:**
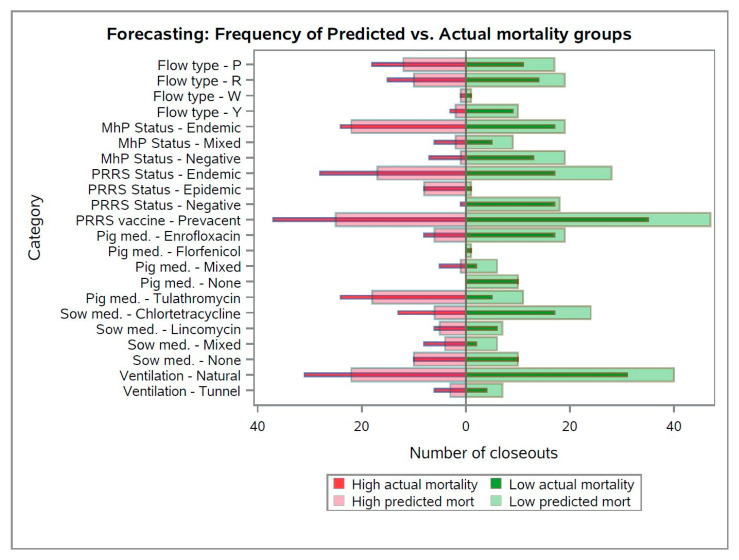
Frequency distribution of the categorical variables included in the forecasting model. Left and right solid bars refers to the number of closeouts with actual high or low nursery mortality (>5%), respectively. Left and right transparent bars refer to the number of closeouts predicted as high or low nursery mortality (<5%), respectively.

**Table 1 animals-13-02412-t001:** Variables selected from the master table for the forecasting analyses.

Data Streams	Variable Type ^‡^	Variables
^(1)^ Breeding Herd Productivity and Health *	Rate	Service repeat rate; Abortion rate; Services per inventory; Proportion of gilts bred; Last week weaned sows bred rate; Proportion of sows pregnant at 105 days; Farrowing rate; Stillborn rate; Mummies rate; Pre-weaning mortality; Pre-natal losses; Sow death rate; Sow culls rate
	Count	Number of services; Number of farrows; Sows inventory
	Average	Wean-to-service interval; Total born; Born alive; Parity at the farrow; Gestation length; Interval between farrows; Pigs weaned/sow; Piglet wean age; Non-productive days; Productive sow days; Litter/female/year; Mated inventory; Pigs/weaned/female/year
^(2)^ Growing Phase Productivity ^†^	Rate	Nursery mortality (mortality on the initial 60 days post placement in a growing site)
^(3)^ Closeouts Health Status *	Category	Group status for porcine reproductive and respiratory syndrome (PRRS) at placement; Group status for *Mycoplasma hyopneumoniae* (MhP) at placement
^(4)^ Pig Transportation *	Time	Weaning movement year; Weaning movement week
	Count	Number of animals transported
^(5)^ Stocking Conditions *	Category	Type of flow; Type of ventilation;
	Count	Number of origins; Time to fill the site; Breeding herd origins
^(6)^ Management Procedure *	Category	Type of PRRS vaccine; Type of piglet medication at weaning; Breeding herd type of mass medication protocol

^(1)–(6)^ Data streams utilized; ^†^ Outcome variable; * Predictor Variables. ^‡^ Type of variables. Variable type classified as “Average” refers to the average number of count events occurring for the batch of groups weaned. For example, “Total Born” variable represents the average number of total piglets born per farrow over a total number of farrowing events in a week. More details about the classes of the categorical variables are described in Table 2.

**Table 2 animals-13-02412-t002:** Description of the categorical variables included in the forecasting model.

Data Type	Variable ^†^	A *	B *	C *	D *	E *	F *
Breeding herd health	PRRS status	Epidemic	Endemic	Mixed	Negative	-	-
	MhP status	Epidemic	Endemic	Mixed	Negative	-	-
Stocking conditions *	Type of flow ^3^	DS-M	DS-S	Y	S	-	-
	Type of ventilation	Tunnel Barn	Curtain Barn	-	-	-	-
Management procedure *	PRRS vaccine	Vaccine A	Vaccine B	-	-	-	-
	Piglet medication ^1^	Enrofloxacin	Tulathromycin	Ceftiofur	Florfenicol	Mixed	None
	Sow medication ^2^	CTC ^4^	Lincomycin	Tilmicosin	Mixed	None	-

* Categories of each variable; ^1^ Type of medication treatment in piglets; ^2^ Type of medication treatment in sows; ^3^ DS-M: Double stock moved; DS-R: Double stock remained; Y: Nursery-to-finisher flows; S: Single stock flows; ^4^ CTC: Chlortetracycline; ^†^ Categorical variables from Table 1.

**Table 3 animals-13-02412-t003:** Performance of the forecasting models on predicting swine nursery mortality.

Model ^1^	Parameters ^2^		
	R^2^	RMSE	MAE
MLR	0.385	0.614	0.475
LASSO	0.392	0.611	0.471
RF	0.725	0.421	0.313
SVM	0.731	0.406	0.284
NNet	0.533	0.566	0.393

^1^ MLR: Multiple Linear Regression; LASSO: LASSO regression; RF: Random Forest; SVM: Support Vector Machine; NNet: Neural Network. ^2^ RMSE: Root Mean Square Error; MAE: Mean Absolute Error; R^2^: r-square.

## Data Availability

The data presented in this study are available on reasonable request from the corresponding author [E.S.M.]. The data are not publicly available due to privacy.

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
