# Peer review of "Field Implementation of Forecasting Models for Predicting Nursery Mortality in a Midwestern US Swine Production System"

_animals, 2023, doi:10.3390/ani13152412_

Round 1

Reviewer 1 Report

The manuscript submission authored by Magalhaes et al. titled “Field implementation of forecasting models for predicting nursery mortality in a Midwestern US swine production system” is highly applicable to swine production globally. The techniques described in this manuscript present a fundamental leap forward in terms of using multiple data streams and aggregating this information together with a clear outcome that can be readily used by swine producers to best allocate resources such as labor and medical intervention. Overall the dataset used is extremely robust and represents a tremendous number of pigs and data capture appears to be very sound. More data does not always equal better data, but in this case the very large quantity of data appears to be collected in a manner to limit potential bias and confounding and is a terrific dataset to use with the forecasting models being developed.

Overall the manuscript is sound and describes the background, methodology, results, and interpretation in sufficient detail and clarity. Please see a few thoughts for consideration to perhaps strengthen the manuscript:

·         When matching sow farm production metrics (wean-service interval, total born, pigs weaned/female/year, how were these metrics matched to a single lot of pigs post-weaning? Was it the sow farm metrics for the entire herd at the time of weaning? With sow farm having 5,000+ sows and a cohort group of weaned pigs only representing a small subset of that sow population, what was the strategy to match these data? A brief description related to this would help the reader understand how these different metrics were aggregated into the master table described.

·         Where pigs from multiple sow farms placed into the same nursery or wean-finish barns? Comingling of multiple sources…did this occur at all, and if so, please provide some detail on how this was handled within the data pipeline. The basic outline of whether a site was single or double stocked, etc, is straight forward, but please provide some more detail related to more complex situations that undoubtedly happen in a commercial production system related to multiple sow farms commingling into a single site, etc, and if that was the case how did the predictive model handle this data when there were multiple sow farms with different production measures?

·         Table 1: Under the breeding herd productive and health section, total born is listed as an average. Should that not be considered count data? There can be 14 total born in a litter, or 15, not 14.5, for example. Same with born alive – could either be count or proportion of total born. Approximating these are normally distributed, continuous data is probably not an unreasonable assumption, but given count data are modeled explicitly for responses such as count of services I would be interested in the authors thoughts on why a similar approach was not used for litter size metrics.

·         Table 1: Under “stocking conditions”, what is meant by “breeding herd origins”?

·         Table 2: This is helpful, perhaps as a footnote in table 1 could reference that table 2 will describe some of the data types in additional detail? I was confused when reading table 1 what was meant by stocking conditions until I read on until table 2. A footnote in table 1 would help avoid this confusion.

·         Table 1: Where is the double cross footnote referenced within the table?

·         Figure 1: The x-axis is not in chronological order. It appears to be in alphabetical order within year. Please update to make the line graph to be in chronological order.

·         Figure 1: the x-axis label of weaning event time is very helpful. I would perhaps make sure this is clear in the figure legend also that time corresponds to the season in which pigs were weaned, not when that closeout finished.

Author Response

Please see the attachment. Thanks for the great review!

Author Response

Please see the attachment. Thanks for your review!

Author Response

(The authors gave the same response as above.)
